# Social anxiety and academic performance during COVID-19 in schoolchildren

Joel Manuel Prieto[1], Jesús Salas Sánchez[1,2]*, Javier Tierno Cordón[1], Leandro Álvarez-Kurogi[1], Higinio González-García[1], Rosario Castro López[1]

1 Universidad Internacional de La Rioja, Logroño, Spain, 2 Universidad Autónoma de Chile, Providencia, Chile

* jesus.salassanchez@unir.net

## Abstract

The purpose of the present study was to determine the perception of schoolchildren whether their academic performance improved or worsened during the pandemic, analyzing their social anxiety, gender, use of masks in the classroom, and school year. The total sample was 107 primary school students (25 in the fourth, 40 in the fifth and 42 in the sixth grade), with a mean age of 10.51 years old (SD = 1). The gender were 58 girls and 49 boys, from a school in the province of La Coruña (Spain). The study was based on a quantitative methodology, and the design was cross-sectional, descriptive, observational and correlational. The social anxiety questionnaire (CASO-N24) was used to assess social anxiety, and an *ad hoc* self-report register was elaborated to evaluate sociodemographic variables. The results indicated that 44.8% of the schoolchildren considered that the pandemic had neither improved nor worsened their academic performance. Although 38.3% considered that high and very high social anxiety increased progressively as the school year progressed, both in boys and girls. Besides, the schoolchildren who presented very low and low social anxiety improved their grades in Physical Education, while those who presented high social anxiety worsened them. In conclusion, having a low social anxiety, lower grades before the pandemic and higher grades after, makes children perceive an improvement in their academic performance during the pandemic.

## Introduction

The health crisis SARS-COV-19 has been considered by the World Health Organization (WHO) as one of the greatest pandemics of the 21st century [1, 2]. The COVID-19 has created an uncertain, sudden and unexpected social, health, educational, economic and personal landscape [1, 2]. In response to the COVID-19 disease, governments around the world obligated the use of facemasks in public spaces to curb transmission of the virus, such as educational facilities [1]. During the pandemic, mental health worsened, obtaining above-average levels of social anxiety [2]. Therefore, the academic performance of children in school population may be seriously impaired by these variables, as it is known that the pandemic has harmed the teaching activity, modifying the way of working [3]. However, it seems that there are few studies, in which it is reflected how mask usage or social anxiety affects the academic performance

**Funding:** Acknowledgments: to the International University of La Rioja, for financing the project with code "PI:024/2022"called "Impact of use of different masks in physical-sports activity in students" in which this research is framed, specifically, from the call for projects "Financing of UNIR Own Projects", as well as to the educational center where the research was carried out.

**Competing interests:** The authors have declared that no competing interests exist.

of children. This is surprising as socialization is crucial to experience adaptive growth in children as well as in academic performance. Hence, this study will further examine the impact of a mask use period in school, after lockdown, and its influence on academic performance and social anxiety.

Masks (hygienic or ffp2) present a visual barrier for those who rely on nonverbal communication cues on the face (e.g., mouth, lips, teeth, tongue and cheeks) [4, 5]. In addition to impairing lip-reading and hearing the voice of teachers and peers more attenuated as well as to cause difficulties in language development [4, 5]. Furthermore, there is increasing evidence that the mask used to prevent the virus in the school environment negatively affects social relationships, communication and emotions [6]. For instance, it seems that the impact of fear and anxiety about a new illness and what might happen in the face of it can be overwhelming and generate strong emotions in adults and children [7]. Recent research has shown that facemasks hinder facial processing skills in adults, including the ability to perceive the face identity [8, 9], their emotional expression [10], and recognize voices [11]. However, it seems that there is little scientific literature that analyzes how the results of this prevention measure affect academic activity in children.

Public health measures, such as social distancing, have caused people to feel isolated and lonely and may increase stress and anxiety [12]. According to Gutiérrez [6] the mask usage is associated with social anxiety, due to it may create an emotional blockage with others because communication does not occur naturally. Thus, creating social rejection and negative feelings of fear, anxiety and phobia. Anxiety in children and adolescents can manifest itself in defiant behavior (e.g., arguing or refusing to obey) [13]. As we can see, there are several studies reflecting pandemic-generated depression and anxiety in adults [14]. However, it seems that social anxiety in children is not fully demonstrated. In this regard, few studies are established in terms of social anxiety in face-to-face classes during the pandemic, focusing the literature mostly on health anxiety [15] or social anxiety in terms of pandemic quarantine [16, 17] or remote learning [18, 19]. Thus, this study will further develop the situation of social anxiety in schoolchildren after the quarantine in a period in which it still been imposed the use of mask in educational centers.

Based on Vigotsky's Sociocultural Theory [20], it is interesting to analyze how the mask usage is affecting this process, since it prevents socialization from occurring naturally, creating a barrier to communication. Moreover, at early age, it is essential the communication due to the importance of vicarious learning in teaching process. Research shows that those who have better academic performance have less anxiety in a statistically significant way [21]. As such, this article will provide insights into scientific literature, due to their measure of academic performance and social anxiety from a different perspective in comparison to the previous literature, because it will be unravelling the gender differences and the school year. This perspective may provide a piece of useful information for teachers and psychologists to be more precise in their strategies in pandemic contexts as well as personalize their teaching strategies to those that are at risk. As such, the aim of the study was to determine the perception of schoolchildren and whether their academic performance improved or worsened during the pandemic, analyzing their social anxiety, gender, use of masks in the classroom, and school year.

## Materials and methods

### Design

The research design was cross-sectional, descriptive and correlational and it was carried out after the lockdown in a period of compulsory mask usage in spanish schools. The dependent variable was social anxiety and the perception of improvement or not on their academic

performance during the pandemic (worsened, neither worsened nor improved, improved). The independent variables were: gender, the influence of the mask usage in the classroom and the school year.

## Participants

The sample is made up of 107 primary school students (25 in the fourth, 40 in the fifth and 42 in the sixth grade), with a mean age of 10.51 years old ($SD$ = 1), 58 girls and 49 boys. The mean of hours studying per week was 6.97 ($SD$ = 7.42). Regarding the participants that suffered the epidemic, 28 had COVID any time and 79 did not suffer the disease. In terms of the sports practice, 93 practiced sports and 14 did not practice any sport. Besides, 49 were federated athletes and 58 were not federated athletes.

## Instrument

The social anxiety questionnaire [22] was used in this study. This measure presents two versions, one aimed at boys and the other at girls, both with 24 items with a Likert-type response scale from 1 to 4, being 1: not at all, 2: a little, 3: quite a lot, and 4: a lot. The items of this questionnaire were obtained from a review of the literature on childhood fears and problematic social situations in children who came for consultation for social phobia. The questionnaire contemplates 6 factors: Public speaking/Interaction with teachers; Interaction with the opposite sex; Being embarrassed or ridiculed; Assertive expression of annoyance or anger; Interaction with strangers; Acting in public. As in this study, the general factor of social anxiety was only taken, the internal consistency reflected in Cronbach's alpha for the entire questionnaire yielded a value of 0.86. In the present study, the sum of the scores obtained in the 24 items of the Social Anxiety questionnaire were used [22], recoding them into different variables, establishing the following categories and cut-off points:

- Very Low Social Anxiety (VLSA: from the lowest score (minimum of 26.00) to the range of 39.58)

- Low Social Anxiety (ASB: range 39.58 to range 53.16)

- High Social Anxiety (ASA: range 53,17 to range 72,09)

- Very High Social Anxiety (ASMA: range 72.10 to the highest score (maximum of 91.00).

## Procedure

The study was approved by the ethics committee of the Universidad Internacional de La Rioja (UNIR), whose approval code is PI024/2022. The study complied with the American Psychology ethical standards [23]. Before beginning with the process of administering the questionnaires, permission was requested from the school center and the parents of the students. In the informed consent, the confidentiality and anonymity of the data were ensured. The questionnaires were administered individually through a Google forms link. The administration of the questionnaires was carried out by teaching and research staff to assist with any doubts that might arise during the procedure.

## Statistical analysis

The SPSS 21.0 statistical package was used. Descriptive analyses and simple linear regressions, by the input method, were performed to establish a model (y = a+bx) to predict

schoolchildren's perception of their academic performance during the pandemic as a function of psychological (social anxiety) and sociodemographic variables (gender, school year and academic performance before and after the pandemic). On the other hand, hypothesis tests were performed, through bivariate correlations, to determine the influence of sociodemographic variables such as gender and school year on social anxiety. All statistical analyses were performed with a significant level of 95%.

## Results

In the total of the schoolchildren surveyed, 44.8% considered that the pandemic had neither improved nor worsened their academic performance, 38.3% considered that it had worsened their academic performance, and 16.8% considered that the pandemic had improved their academic performance. Table 1 shows how the proportion of boys consider that the pandemic has worsened their academic performance (57.14%) compared to girls (22.41%), while the majority of girls consider that it has neither worsened nor improved their academic performance (56.89%).

On the other hand, gender and school year were correlated with the influence of mask use on concentration and learning, with no significant results ($p > .05$). Analyzing frequencies, 32.7% considered that the mask use in the classroom did not influence their concentration and learning, 16.8% that it influenced, 27.10% thought it had a great influence, 11.2% that worsened and 12.4% that their concentration and learning worsened a great deal with the mask usage in the classroom. Regarding the social anxiety of the surveyed sample, 45.80% had high social anxiety, 35.51% had low social anxiety, 12.15% had very low social anxiety, and 6.54% had very high social anxiety. Fig 1 shows the percentages of social anxiety in boys and girls.

Fig 2 shows the values of social anxiety by school year, showing that High and Very High Social Anxiety increase progressively from fourth to fifth grade and from fifth to sixth grade. The correlation between both variables allows to determine that the strength of association is adequate ($p < .05$).

Regarding the relationship between academic performance and social anxiety, the *ad hoc* questionnaire asked them to indicate their score before and after the pandemic in 6 subjects: natural sciences, mathematics, social sciences, Spanish language, English and physical education. There was only a significant relationship between social anxiety and academic performance before and after the pandemic for the subject of physical education ($p < .05$). Schoolchildren who presented very low (8.15/8.54) and low (8.39/8.63) social anxiety improved their grades in Physical Education, while those who presented high social anxiety (8.35/8.27), worsened them. On the other hand, there were no significant relationships between social anxiety and academic performance by gender or school year. Two new variables were calculated based on the sum of grades before and after the pandemic, and there were no significant relationships between social anxiety and academic performance by gender or school year.

**Table 1. Influence of the pandemic on academic performance.**

|  | Influence of the use of the mask | Count |
|---|---|---|
|  | Has worsened | 13 |
| Girls | It has neither worsened nor improved | 33 |
|  | Improved | 12 |
|  | Has worsened | 28 |
| Boys | It has neither worsened nor improved | 15 |
|  | Improved | 6 |

Note. Correlation $p = 0.01$ following Kendall's tau b test.

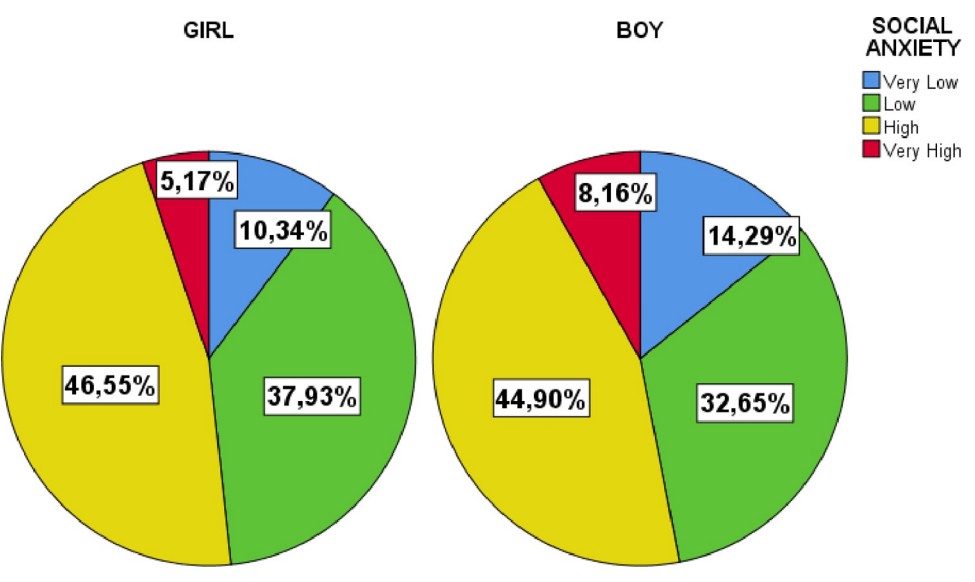

**Fig 1. Social anxiety in boys and girls.**

Table 2 shows a linear regression, which relates, as a dependent variable, the improvement or not of academic performance during the pandemic, with gender, school year, social anxiety and academic performance before and after the pandemic. Table 2 shows that the variables that contribute most to the variance (52.3%) are gender, performance before the pandemic, performance after the pandemic, and social anxiety. A total of 23.7% of the variance can be predicted by all the factors listed in Table 2. The combination of these variables significantly predicts the improvement or not of academic performance during the pandemic ($F(7.591) = 14.76$, $p < .001$). The

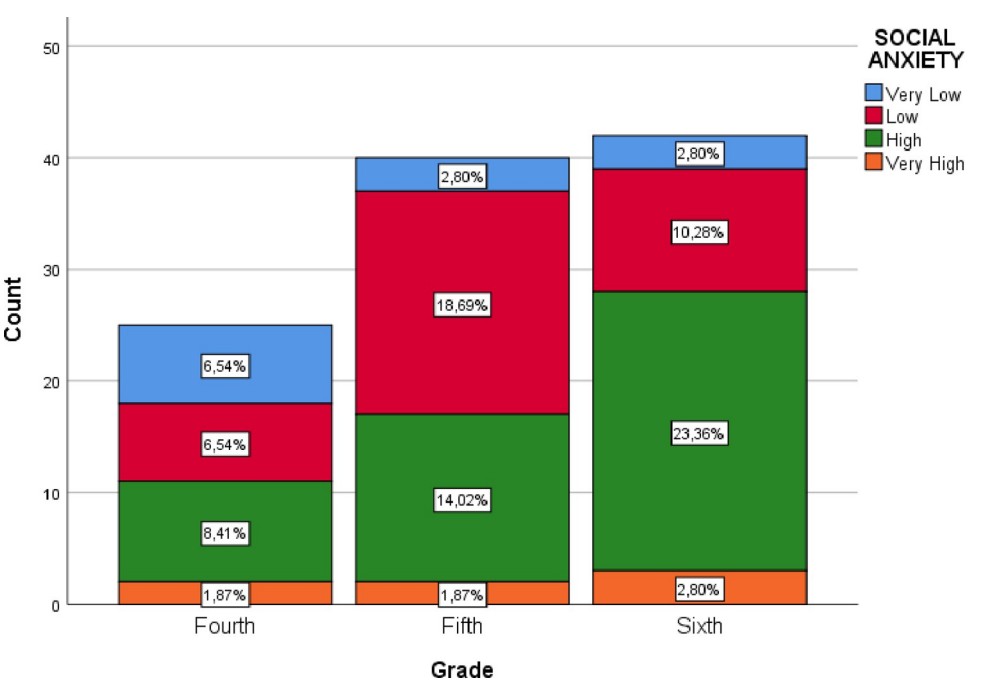

**Fig 2. Social anxiety by school year.**

**Table 2. Linear regression for improvement or worsening of academic performance during the pandemic.**

| Step | Predictor variables | | $\beta$ | $p$ | $R^2$ | $p$ |
|------|---------------------|--|---------|-----|-------|-----|
| 1 | **Personal, psychological and academic** | | | | | |
| | | Gender | -0.363 | 0.004 | | |
| | | Course | -0.129 | 0.110 | | |
| | | Pre-pandemic performance | -0.046 | 0.001 | | |
| | | Post-pandemic performance | 0.037 | 0.002 | | |
| | | Social Anxiety | -0.205 | 0.010 | | |
| | | | | | 52.3% | < .001 |
| | | | | $R^2$ | 27.3% | < .001 |
| | | | | $R^2_{adj.}$ | 23.7% | < .001 |

model (y = a+x) whose dependent variable (y) is the improvement or not of academic performance during the pandemic, has a constant (a) of 3.14, with the coefficient (as indicated in Table 2 for each factor).

Table 1 shows that 56.89% of the girls considered that their academic performance neither worsened nor improved, with significance, as shown in Table 1 ($\beta$ = -0.36; $p$ < .01). On the other hand, it can be seen that a lower academic performance before the pandemic influences their perception of improvement in their academic performance during the pandemic ($\beta$ = -0.04; $p$ < .001). Otherwise, a higher academic performance after the pandemic also influences their perception about the improvement in their academic performance during the pandemic ($\beta$ = -0.03; $p$ < .01). Indeed, schoolchildren who perceived that they got worse had a lower score after the pandemic compared to their previous score (7.78/7.42), those who perceived that they neither got worse nor better had a similar score (7.70/7.69), and those who perceived that they got better during the pandemic had a better score (7.02/7.49). On the other hand, lower social anxiety influenced their perception of improved academic performance during the pandemic (improved: 47.06; neither improved nor worsened: 53.00; worsened: 56.02) ($\beta$ = -0.20; $p$ < .01). The school year had no significant relationship with the perception of schoolchildren on the improvement or not of academic performance during the pandemic.

## Discussion

The study aimed to determine the perception of schoolchildren whether their academic performance improved or worsened during the pandemic, analyzing their social anxiety, gender, use of masks in the classroom, and school year. In this regard, a large percentage of the sample of schoolchildren (38.3%) considered that their academic performance has worsened as a result of the pandemic caused by COVID-19, and that high and very high social anxiety increases progressively as the school year progresses. It is also noteworthy that schoolchildren with high social anxiety had worse overall grades in Physical Education. The subject of Physical education is very participatory and implies that the students are in motion and in constant organizational changes. Thus, the relationship with others and socialization is fundamental for the integral development of the students. Probably, the social distancing measures prevented the correct affective, emotional and social development of the students, which would cause a decrease in their academic performance. Due to the great phenomenon that has occurred in recent years regarding the incursion of COVID-19 in our society, many studies try to clarify the reality from different perspectives. Specifically, regarding the area of education, it highlights studies such as the one proposed by Chávez and Márquez [24], which states that

university students are a sector of the population that has been severely impacted, manifesting high anxiety states. This result coincides with that proposed by Sigüenza and Vílchez [25] who detected that the final anxiety of university students during the COVID-19 pandemic increased 2.59 points, in relation to the pre-test mean before lockdown. Regarding the school environment, Estacio [26] pointed out that school anxiety presents negative relationships with social self-concept, as well as with emotional self-concept. One of the explanations may be due to the use of masks since the masks usage in the school environment negatively affects social relations, communication or emotions [6].

Considering as a starting point the scientific literature reviewed, shows that 52.34% of schoolchildren from this sample present high and very high levels of social anxiety, a very significant data which provides information on factors such as speaking in public or interacting with teachers; interacting with the opposite sex; being in evidence or ridicule; assertive expression of annoyance or anger; interacting with strangers or acting in public. All these aspects have been impaired by the pandemic situation, which coincides with several studies reviewed in which the deterioration of emotional stability in the population worldwide is evident, adding to the factors that impact the physical and mental health of people after the time lived in lockdown [27–29]. Some studies suggest the inclusion in the curriculum of emotional education to address and counteract social anxiety in order to provide children, youth, parents and teachers with skills, attitudes and behaviors necessary to stay healthy and positive, explore their emotions, practice mindful engagement, exhibit prosocial behavior and deal with daily challenges [30, 31].

In fact, this study supports that high social anxiety is related to lower overall Physical Education scores than the rest of the participants, which indicates once again that physical activity, regardless of the global health crisis, remains closely related to psychological well-being. Since decades, scientific literature has demonstrated a significant decrease in symptoms in different anxiety disorders through the implementation of exercise programs [32], specifying that higher levels of PA (Physical Activity) correspond to lower levels of anxiety and higher well-being even during the most intense time of COVID-19 epidemic in China [33, 34]. These studies, together with our finding, coincide in pointing out the impact of PA on anxiety levels generated by COVID-19, which should be recommended to help or improve the handle of anxiety and stress experienced during the COVID-19 pandemic, as there is also a clear prediction of the negative effect on long-term psychological health [28, 35].

Another objective was to analyze the students' perception of their academic performance, understanding that, to some extent, it could be affected. In this sense, a total of 38.3% of the sample considers that it has worsened, an assumption that coincides with the study [36], which indicates that virtuality, mood and family coexistence affected the academic performance of students. Only certain measures to curb the pandemic can lead to a detriment of academic performance, such as wearing the mask [37], evidencing that wearing it in Physical Education classes can influence hypoxic and hypercapnic breathing at a certain intensity [38]. There are no studies comparing performance before and after the pandemic, but as reflected in the previous data, affective, social, emotional aspects or wearing a mask make academic performance during the pandemic to be lower. In this line, students with best middle school marks have less social anxiety to avoid interacting with strangers [39].

Regarding the limitations of the study, we can mention the small number of subjects in the study sample. On the other hand, it is not measure data on the different types of masks that students may have used in the classroom during the pandemic. This could hinder the applicability of the results to a certain type of mask. As such, it would be interesting to examine how each type of mask may mediate the relationship with academic performance. Regarding methodological procedures, the lack of evaluation of other physiological or psychological measures

would provide a more complete picture of the results. For instance, it would be interesting to examine the impact of teachers on academic performance as they have changed their teaching situation in the epidemics [40]. On the other hand, the use of self-report measures may lead to various biases such as social desirability, acquiescence, etc. Nevertheless, the psychometric properties of the instruments used were appropriate, which strengthens the quality of the results found.

The present study provides results that increase the theoretical foundation on social anxiety and its relationship with academic performance, being very useful to better understand the relationship between these aspects. Taking into account that these results can only be extrapolated to the sample of this study focused on Primary Education, it is possible that subjects of higher ages have different perceptions of their performance and levels of social anxiety than those analyzed in this work.

In future research, we propose the development of similar predictive models that help to observe the different factors that influence on the academic performance of schoolchildren in periods of lockdown or pandemic. The probability of experiencing social anxiety or other psychopathologies in young people is associated with different factors, such as wearing a mask, therefore, it is proposed continue research in this line.

## Conclusions

The aim of this study was to determine the perception of schoolchildren if their academic performance improved or worsened during the pandemic, analyzing as influential variables their social anxiety, gender, the use of masks in the classroom, and the school year. The main findings are listed below:

- 44.8% of schoolchildren consider that the pandemic has neither improved nor worsened their academic performance, although 38.3% consider that it has worsened it, with a higher proportion of boys considering that the pandemic has worsened their academic performance (57.14%) compared to girls (22.41%).

- High and very high social anxiety increases progressively as the school year progresses, both in boys and girls.

- Regarding to academic performance before and after the pandemic, schoolchildren with very low and low social anxiety improved their grades in Physical Education, while those with high social anxiety worsened their grades.

- Possessing low social anxiety, having had lower grades before the pandemic and higher grades after the pandemic, makes them perceive an improvement in their academic performance during the pandemic.

- There were no significant relationships between gender and school grade with the influence of mask use on concentration and learning in schoolchildren. Likewise, there were no significant relationships between social anxiety and academic performance by gender or school year.

## Author Contributions

**Conceptualization:** Joel Manuel Prieto, Jesús Salas Sánchez, Javier Tierno Cordón, Leandro Álvarez-Kurogi, Higinio González-García, Rosario Castro López.

**Data curation:** Joel Manuel Prieto, Jesús Salas Sánchez, Javier Tierno Cordón, Leandro Álvarez-Kurogi, Higinio González-García, Rosario Castro López.

**Formal analysis:** Joel Manuel Prieto, Jesús Salas Sánchez, Javier Tierno Cordón, Leandro Álvarez-Kurogi, Higinio González-García, Rosario Castro López.

**Funding acquisition:** Joel Manuel Prieto, Jesús Salas Sánchez, Javier Tierno Cordón, Leandro Álvarez-Kurogi, Higinio González-García, Rosario Castro López.

**Investigation:** Joel Manuel Prieto, Jesús Salas Sánchez, Javier Tierno Cordón, Leandro Álvarez-Kurogi, Higinio González-García, Rosario Castro López.

**Methodology:** Joel Manuel Prieto, Jesús Salas Sánchez, Javier Tierno Cordón, Leandro Álvarez-Kurogi, Higinio González-García, Rosario Castro López.

**Project administration:** Joel Manuel Prieto, Jesús Salas Sánchez, Javier Tierno Cordón, Leandro Álvarez-Kurogi, Higinio González-García, Rosario Castro López.

**Resources:** Joel Manuel Prieto, Jesús Salas Sánchez, Javier Tierno Cordón, Leandro Álvarez-Kurogi, Higinio González-García, Rosario Castro López.

**Software:** Joel Manuel Prieto, Jesús Salas Sánchez, Javier Tierno Cordón, Leandro Álvarez-Kurogi, Higinio González-García, Rosario Castro López.

**Supervision:** Joel Manuel Prieto, Jesús Salas Sánchez, Javier Tierno Cordón, Leandro Álvarez-Kurogi, Higinio González-García, Rosario Castro López.

**Validation:** Joel Manuel Prieto, Jesús Salas Sánchez, Javier Tierno Cordón, Leandro Álvarez-Kurogi, Higinio González-García, Rosario Castro López.

**Visualization:** Joel Manuel Prieto, Jesús Salas Sánchez, Javier Tierno Cordón, Leandro Álvarez-Kurogi, Higinio González-García, Rosario Castro López.

**Writing – original draft:** Joel Manuel Prieto, Jesús Salas Sánchez, Javier Tierno Cordón, Leandro Álvarez-Kurogi, Higinio González-García, Rosario Castro López.

**Writing – review & editing:** Joel Manuel Prieto, Jesús Salas Sánchez, Javier Tierno Cordón, Leandro Álvarez-Kurogi, Higinio González-García, Rosario Castro López.

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
