## [Decision Letter · Decision Letter 0]

1 Dec 2022

PONE-D-22-24450Social anxiety and academic performance during covid-19 according to gender, school year and use or non-use of masksPLOS ONE

Dear Dr. Salas Sánchez,

Thank you for submitting your manuscript to PLOS ONE. After careful consideration, we feel that it has merit but does not fully meet PLOS ONE’s publication criteria as it currently stands. Therefore, we invite you to submit a revised version of the manuscript that addresses the points raised during the review process.

We look forward to receiving your revised manuscript.

Kind regards,

Ender Senel, PhD

Academic Editor

PLOS ONE

Journal Requirements:

a) Did participants provide their written or verbal informed consent to participate in this study?

Acknowledgments: to the International University of La Rioja, for financing the project in which this research is framed, specifically, from the call for projects “Financing of UNIR Own Projects”, as well as to the educational center where the research was carried out

Reviewers' comments:

Reviewer's Responses to Questions

**Comments to the Author**

1. Is the manuscript technically sound, and do the data support the conclusions?

Reviewer #1: Partly

Reviewer #2: Yes

Reviewer #3: Yes

2. Has the statistical analysis been performed appropriately and rigorously? 

Reviewer #1: N/A

Reviewer #2: Yes

Reviewer #3: Yes

3. Have the authors made all data underlying the findings in their manuscript fully available?

Reviewer #1: Yes

Reviewer #2: Yes

Reviewer #3: Yes

4. Is the manuscript presented in an intelligible fashion and written in standard English?

Reviewer #1: No

Reviewer #2: Yes

Reviewer #3: Yes

5. Review Comments to the Author

Reviewer #1: I think you have problems with timeliness and originality.When planning special works of ancient times, you evaluate the timeliness of your originality and originality qualities. Considering the Kovid effect, the use of masks and other vital processes could have made the study valuable. This version of the work is also valuable and meaningful, but I think it does not have an international level of quality and content.

Reviewer #2: Dear Authors,

Thank you very much for providing me the opportunity to read such an interesting article.

However, some concerns should be addressed before publication. See them here listed below:

- The title should be shortened according to the international writing guidelines.

- Lines 12-14. The phrase could be shortened to be more dynamic in reading.

- Line 71. There are two redundancies of the word "From".

- In the introduction. it is stated that there is a lack of literature on academic performance in COVID-19, but the article does not explain it. Although there is a lack of articles, it is needed to explain what is done and what may add this work as a plusvalue.

- Lines 101-104. It is needed to base this classification on previous authors.

- In table 1. It seems that there is missing information.

- Line 208. Please check the phrase.

- Line 218. Here you can say the surnames of the authors. Please check the Vancouver style to be more precise.

- Line 267-268 and 284. Please check the phrase to be grammatically correct.

- Line 297. The linker utilized is not the most suitable one. Please check.

Reviewer #3: -On page 9, Table 2 discusses sports and personal predictor variables. The reference to "sports variables" should be removed.

-Add sources in the introduction that show the relationship between social anxiety and academic performance.

-Argue in the first paragraph of the discussion why schoolchildren with high social anxiety worsened their overall grades in Physical Education.

-Delete on page 12 the names of citation 36: Miranda, Margoth, Pantoja and Valdivieso

6. PLOS authors have the option to publish the peer review history of their article (what does this mean?). If published, this will include your full peer review and any attached files.

Reviewer #1: No

Reviewer #2: **Yes: **Alfonso Castillo Rodríguez

Reviewer #3: **Yes: **ALFONSO TRINIDAD MORALES

---

## [Author Response · Author response to Decision Letter 0]

16 Dec 2022

Dear Reviewers,

Thanks for your suggestions in the improvement of the manuscript. The recommendations of the three reviewers have been followed. According to that, below you can see a list of the corrections that have been made:

Journal Requirements: 

The participants gave their written informed consent. The document from line 123 to 128 talks about it.

The performance of the funders in the study has been included in the cover letter.

Reviewer 1

- To be more precise in the valuable insights of this study it was added in lines 39-42 the surplus of the study as well as in the last paragraph of the introduction.

- In lines 43- 67 it was stated the current state of art in mask usage and their impact.

- Considering the COVID effect it was added the period in which the study was carried out in the introduction. See lines 67-69. 

- In addition, in the design section it was clarified the period in which the study was carried out to highlight that it was after lockdown and in a compulsory mask usage.

Reviewer 2

-The title was shortened according to the suggestion. Thanks so much.

- Lines 12-14. The phrase was shortened to be more precise.

- Line 71. It was used the word “in comparison to” to do not be redundant.

- In the introduction. It was added the surplus of the study in the lines 39-40 as well as in the last paragraph of the introduction.

- Lines 111-112. It was cited that the classification taken was from the following study:

22. Caballo VE, Arias B, Salazar IC, Calderero M, Irurtia MJ, Ollendick TH. Un nueva medida de autoinforme para evaluar la ansiedad/fobia social en niños: el "Cuestionario de ansiedad social para niños"(CASO-N24). Behavioral Psychology/Psicología Conductual. 2012;20(3):485-504.

- The table 1 was completed according to the suggestions made.

- The manuscript was revised to correct typos and to abide by the Vancouver style.

Reviewer 3

-The reference to "sports variables" in Table 2 has been removed.

-A quotation has been added in the introduction that show the relationship between social anxiety and academic performance.

-In the first paragraph of the discussion (218-223) has been discussed why schoolchildren with high social anxiety worsened their overall grades in Physical Education.

-The names of citation 36: Miranda, Margoth, Pantoja and Valdivieso have been removed.

---

## [Editor Report · Decision Letter 1]

22 Dec 2022

SOCIAL ANXIETY AND ACADEMIC PERFORMANCE DURING COVID-19 IN SCHOOLCHILDREN

PONE-D-22-24450R1

Dear Dr. Salas Sánchez,

We’re pleased to inform you that your manuscript has been judged scientifically suitable for publication and will be formally accepted for publication once it meets all outstanding technical requirements.

Kind regards,

Ender Senel, PhD

Academic Editor

PLOS ONE

---

## [Editor Report · Acceptance letter]

3 Jan 2023

PONE-D-22-24450R1 

Social anxiety and academic performance during COVID-19 in schoolchildren 

Dear Dr. Salas Sánchez:

I'm pleased to inform you that your manuscript has been deemed suitable for publication in PLOS ONE. Congratulations! Your manuscript is now with our production department. 

Kind regards, 

on behalf of

Dr. Ender Senel 

Academic Editor

PLOS ONE